# Dysregulated activities of proline-specific enzymes in septic shock patients (sepsis-2)

**Gwendolyn Vliegen**[1], **Kaat Kehoe**[1], **An Bracke**[1], **Emilie De Hert**[1], **Robert Verkerk**[1], **Erik Fransen**[2], **Bart 's Jongers**[3], **Esther Peters**[4], **Anne-Marie Lambeir**[1], **Samir Kumar-Singh**[3], **Peter Pickkers**[4], **Philippe G. Jorens**[5], **Ingrid De Meester**[1] *

**1** Laboratory of Medical Biochemistry, University of Antwerp, Antwerp, Belgium, **2** StatUa Center for Statistics, University of Antwerp, Antwerp, Belgium, **3** Molecular Pathology Group, Cell Biology and Histology, University of Antwerp, Antwerp, Belgium, **4** Department of Intensive Care Medicine, Radboud university medical center, Nijmegen, The Netherlands, **5** Department of Critical Care Medicine, Antwerp University Hospital, Edegem, Belgium and Laboratory of Experimental Medicine and Pediatrics, University of Antwerp, Antwerp, Belgium

* ingrid.demeester@uantwerpen.be

**Data Availability Statement:** The SPSS file containing all raw data is available from Figshare (DOI 10.6084/m9.figshare.10252877.). Processed

## Abstract

The proline-specific enzymes dipeptidyl peptidase 4 (DPP4), prolylcarboxypeptidase (PRCP), fibroblast activation protein α (FAP) and prolyl oligopeptidase (PREP) are known for their involvement in the immune system and blood pressure regulation. Only very limited information is currently available on their enzymatic activity and possible involvement in patients with sepsis and septic-shock. The activity of the enzymes was measured in EDTA-plasma of patients admitted to the intensive care unit (ICU): 40 septic shock patients (sepsis-2) and 22 ICU control patients after major intracranial surgery. These data were used to generate receiver operating characteristic (ROC) curves. A survival analysis (at 90 days) and an association study with other parameters was performed. PRCP (day 1) and PREP (all days) enzymatic activities were higher in septic shock patients compared to controls. In contrast, FAP and DPP4 were lower in these patients on all studied time points. Since large differences were found, ROC curves were generated and these yielded area under the curve (AUC) values for PREP, FAP and DPP4 of 0.88 (CI: 0.80–0.96), 0.94 (CI: 0.89–0.99) and 0.86 (CI: 0.77–0.95), respectively. PRCP had a lower predicting value with an AUC of 0.71 (CI: 0.58–0.83). A nominally significant association was observed between survival and the DPP4 enzymatic activity at day 1 (p<0.05), with a higher DPP4 activity being associated with an increase in survival. All four enzymes were dysregulated in septic shock patients. DPP4, FAP and PREP are good in discriminating between septic shock patients and ICU controls and should be further explored to see whether they are already dysregulated in earlier stages, opening perspectives for their further investigation as biomarkers in sepsis. DPP4 also shows potential as a prognostic biomarker. Additionally, the associations found warrant further research.

data are within the manuscript and its Supporting Information Files.

**Funding:** EDH and BSJ are research assistants of the Fund for Scientific Research Flanders (1S22417N and SB151525, respectively; www.fwo. be). This work was supported by a GOA BOF 2015 grant of the University of Antwerp to SKS, AML and IDM (No. 30729; www.uantwerp.be). The funders had no role in study design, data collection and analysis, decision to publish, or preparation of the manuscript.

**Competing interests:** The authors have declared that no competing interests exist.

**Abbreviations:** APACHE II, Acute Physiology and Chronic Health Evaluation II; AUC, area under the curve; CI, confidence interval; DPP4, dipeptidyl peptidase 4; DTT, dithiothreitol; EDTA, ethylenediaminetetraacetic acid; FAP, fibroblast activation protein α; FGF21, fibroblast growth factor 21; Gly-Pro-*p*NA, glycyl-prolyl-*para*nitroanilide; ICU, intensive care unit; I-FABP, intestinal fatty-acid binding protein; IFNγ, interferon γ; IL, interleukin; IL-1RA, interleukin-1 receptor antagonist; IQR, interquartile range; MAP, mean arterial pressure; NA, not applicable; PaO$_2$/FiO$_2$ ratio, ratio of arterial oxygen partial pressure to fractional inspired oxygen; PRCP, prolylcarboxypeptidase; PREP, prolyl oligopeptidase; ROC, receiver operating characteristic; SIRS, systemic inflammatory response syndrome; TNFα, tumor necrosis factor α; U/L, units per liter; Z-Gly-Pro-AMC, N-benzyloxycarbonyl-Gly-Pro-7-amido-4-methylcoumarine; Z-Pro-Phe, N-benzyloxycarbonyl-Pro-Phe.

# 1. Introduction

Sepsis remains a major problem in the intensive care unit (ICU), where approximately one third of admitted patients is diagnosed with sepsis [1]. Sepsis is currently defined as a life-threatening organ dysfunction caused by a dysregulated host response to infection and septic shock is defined as a subset of sepsis in which particularly profound circulatory, cellular, and metabolic abnormalities are associated with a greater risk of mortality than with sepsis alone [2]. It is now acknowledged that sepsis involves both pro- and anti-inflammatory responses in combination with alterations in other immunologic and non-immunologic pathways [2–5]. To date, there is no standard diagnostic test that can identify patients with sepsis accurately, although early identification is necessary to improve their outcome [6].

The proline-specific peptidases studied herein are dipeptidyl peptidase 4 (DPP4; EC 3.4.14.5), prolylcarboxypeptidase (PRCP; EC 3.4.16.2), fibroblast activation protein α (FAP; 3.4.21.B28) and prolyl oligopeptidase (PREP; EC 3.4.21.26). All four enzymes are proline-specific preferentially cleaving peptides after proline residues. However, the exact position of the proline in the peptide and the preferred substrates differ between the enzymes. In addition, they share structural properties and therefore belong to related peptidase families, namely S9 (PREP, FAP and DPP4) and S28 (PRCP) [7,8]. Here, we studied the enzymatic activities of DPP4, PRCP, FAP and PREP in the plasma of patients with septic shock (sepsis-2), since they are known for their involvement in the immune system and blood pressure regulation, important factors playing a key role in sepsis, as will be discussed below. Additionally, they are easily measured in plasma or serum with specific assays. Apart from one study on DPP4 activity in severe sepsis patients, the activity of these enzymes in sepsis patients have not been reported in the past. However, it is conceivable that they could be involved in pathogenesis of sepsis and septic shock, as will be outlined below.

DPP4 activity has been shown to be decreased in the serum of patients with sepsis compared to healthy controls [9]. DPP4, also known as CD26, is a co-stimulatory molecule for T-cells and has several substrates that can be involved in sepsis. Examples are neuropeptide Y [10], glucagon-like peptide-1 [11–13], procalcitonin [14,15] and several chemokines [16]. A nested case-control study in type 2 diabetes patients admitted for sepsis did not find a significant association between the use of a DPP4-inhibitor and the development of sepsis [17]. This is reassuring from the therapeutic point of view but tells us nothing about a possible value of DPP4 activity in plasma as a biomarker for sepsis.

FAP, a structurally related ectopeptidase, is, under normal conditions, mostly absent from adult tissues. However, a soluble form can be found in the blood [18]. In a baboon model of sepsis, gene expression of FAP in the lung was maximal at 24 hours post *E. coli* challenge, probably reflecting the active tissue remodeling [19]. FAP also cleaves several substrates, such as α$_2$-antiplasmin and fibroblast growth factor (FGF) 21. Interfering with the cleavage of α$_2$-antiplasmin by FAP might enhance its thrombolytic activity [20] and it has been reported to be increased in patients with septic melioidosis compared to healthy controls [21]. FGF21 has been shown to be increased in patients with sepsis compared to healthy controls and decreased with clinical improvement [22].

The third enzyme studied here is PRCP, Erdös and co-workers reported an increased PRCP activity in the plasma of dogs 20 minutes after endotoxin injection [23]. PRCP has a dual position in the kallikrein-kinin and renin-angiotensin system, by activating prekallikrein and hydrolyzing angiotensin II to form angiotensin (1–7) [24–26]. PRCP is involved in the regulation of blood pressure and hypotension is common in sepsis and septic shock, moreover, angiotensin II is recently approved as therapy for distributive shock. Therefore, a possible role for PRCP in the pathogenesis of sepsis and septic shock is conceivable [4,27,28].

To the best of our knowledge, there are no reports on PREP in sepsis or septic shock, but since it shares several substrates with PRCP, it could be implicated in sepsis and septic shock [29–32]. In addition, PREP shares structural properties and substrate specificities with the other proline-specific peptidases and is therefore included in the study.

The goals of this study were first to evaluate the activities of these enzymes in patients with septic shock (sepsis-2). Since large differences between the septic shock and the ICU control patients were found, ROC curves were generated to test whether these enzymes should be further explored as diagnostic biomarkers. Secondly, the potential of these enzymes as prognostic biomarkers of survival at day 90 in septic shock was evaluated. By exploring associations between the enzymes and a variety of inflammatory, hemodynamic, metabolic parameters, measured on the same days, we additionally aimed to deepen our insights in their possible involvement in septic shock.

## 2. Materials and methods

### 2.1. Patient samples

This prospective cohort study was conducted in patients with septic shock ($\geq$ 18 years) admitted to the ICU of the Radboud University Medical Center (Radboudumc, Nijmegen, the Netherlands). Septic shock was defined according to the definitions stated by ACCP and SCCM consensus conference: a suspected infection, two or more systemic inflammatory response syndrome (SIRS) criteria and the need for vasopressor therapy (Sepsis-2) [33]. All patients received standard of care according to the surviving sepsis campaign guidelines [6,34]. A group of 40 phenotypically well-characterized patients in whom samples were available at all 4 time points (days 1, 3, 5 and 7 after diagnosis) was used. The non-septic shock ICU control group consisted of 22 consecutively admitted patients ($\geq$ 18 years) undergoing major intracranial surgery (resection of a cerebral tumor or clipping of an aneurysm) who were admitted to the ICU of the Antwerp University Hospital for postoperative monitoring. Collection of samples from septic shock patients was done in accordance with the applicable rules concerning the review of the Ethics Committee of UMC Radboud (CMO-nr: 2016/2923) and informed consent was given by the patient or his/her closest relative. All ICU control patients gave written informed consent and the study was approved by the Ethics Committee of the Antwerp University Hospital/ University of Antwerp (Amendment 17/10/119 ref. B300201732219). The study was performed in accordance with the Declaration of Helsinki. Additionally, 30 healthy controls were included, more information on this study group can be found in S1 Appendix and S1 Fig.

### 2.2. Blood sampling

Ethylenediaminetetraacetic acid (EDTA) and lithium heparin-anticoagulated blood from septic shock patients was collected from the arterial catheter within 24 h after diagnosis of septic shock (day 1), and on days 3, 5 and 7. Blood was centrifuged (1600 x g, 4 ˚C, 10 min) and the plasma was stored at -80 ˚C until further analysis. Demographic, clinical and laboratory data were collected on the days of blood sampling. EDTA-plasma from the ICU control group was collected the morning after surgery. All samples from the different groups were handled and stored identically as in the septic shock group.

### 2.3. Enzyme activity measurements

Enzyme activities were measured using in-house validated assays in EDTA-plasma samples. The enzyme activities are expressed in units per liter (U/L) where one unit defines the amount

of enzyme that hydrolyses 1 μmol of substrate per minute under the given assay conditions. Buffers were pH adjusted at room temperature.

DPP4 activity was measured colorimetrically using the substrate glycyl-prolyl-*para*nitroanilide (Gly-Pro-*p*NA; Bachem, Bubendorf, Switzerland) as described earlier [35]. The release of *p*NA from the substrate was measured kinetically at 405 nm during 10 minutes at 37 °C, pH 8.3, using the Infinite™ 200 (Tecan Trading AG, Switzerland).

FAP activities were measured using N-benzyloxycarbonyl-Gly-Pro-7-amido-4-methylcoumarine (Z-Gly-Pro-AMC; Bachem, Bubendorf, Switzerland) as a substrate. Samples were diluted 26 times in a 0.1 M Tris buffer pH 8.0 containing 300 mM NaF, 1 mM NaN$_3$, 1 mM EDTA and 50 mM salicylic acid. 6 μL of 4.6 mM substrate in methanol was added to 150 μL of diluted sample and incubated for 2 h at 37 °C. Reactions were stopped with 500 μL of 1.5 M acetic acid and fluorescence was measured ($\lambda_{ex}$ = 370 nm, $\lambda_{em}$ = 440 nm) on the Shimadzu Fluorimeter RF-5000 (Den Bosch, The Netherlands) [36,37]. Concentration of the generated AMC was determined by means of a standard curve.

A reversed-phase high-performance liquid chromatography technique was used to measure the PRCP activity in the samples as described earlier [38]. PRCP activity was determined by measuring the hydrolysis of N-benzyloxycarbonyl-Pro-Phe (Z-Pro-Phe; Bachem, Bubendorf, Switzerland). Plasma samples were incubated for 2 h with Z-Pro-Phe at 37 °C, subsequently stop solution (10% perchloric acid and 20% acetonitrile solution in purified water (v/v)) was added to stop the enzymatic reaction. The enzymatically formed Z-Pro was tracked by its UV absorbance at 210 nm after separation on a Shimadzu HPLC apparatus equipped with a Chromolith C18e 100x3 mm column, a LC-20AT pump, a SIL-20AC HT autosampler and SPD-20 UV-VIS detector (Shimadzu, Den Bosch, The Netherlands). Quantification was performed by peak height measurements.

PREP in plasma was measured after activation with dithiothreitol (DTT) using the fluorogenic substrate Z-Gly-Pro-AMC as described before with some modifications [37,39]. To 20 μL of plasma, 100 μL of activation buffer containing 100 mM potassium phosphate buffer pH 8.0, 1 mM NaN$_3$, 1 mM EDTA and 10 mM DTT was added. As a blank, 100 μL of the same buffer without DTT was added to 20 μL of sample. 6 μL of 4.6 mM substrate in dimethyl sulfoxide was added to the blank and activation sample tubes followed by incubation for 2 h at 37 °C. The reaction was stopped with 500 μL of 1.5 M acetic acid and fluorescence was measured as described above for FAP activity.

### 2.4. Statistical analysis

Data are represented as median with the interquartile range (IQR), unless otherwise stated.

**Comparison of enzyme activities.** The comparison of enzyme activities between the control group and the different time points in the cases group was carried out using a linear mixed model with 'day' as fixed effect having five levels (day 1, 3, 5, 7 and controls). A random intercept term for individual was added, to account for the non-independence between observations from the same individual. To compare the control level with the four measurements in the patients, a post-hoc analysis was carried out with Dunnett's correction for multiple testing. In addition, changes in mean enzymatic activity between the different days were tested for significance in the patients, using a post-hoc analysis with Tukey's correction for multiple comparisons. For PREP, the test was carried out on the log-transformed enzyme activity due to the non-normality of the residuals.

**Receiver operating characteristic (ROC) curves.** ROC curves were computed to assess the four enzymes as possible biomarkers for septic shock. For the septic shock patients (n = 40) only day 1 was used. The patient's condition on day 1 is clinically the most relevant,

since it is of utmost importance to identify patients as soon as possible to improve their outcome [6]. ROC curves were generated using the ICU control group. Cutoff values were determined using two methods, one in which equal weight is given to false positives and false negatives, also called the Youden index, and another one in which all patients with sepsis are identified, meaning that the sensitivity reaches 100%. To assess combinations of enzymes as predictors of septic shock, the two enzymes were entered as independent variables into a logistic regression model with disease status as dependent variable. Subsequently, the predicted probability of this model was used as an input for the ROC-curve upon which the AUC calculation was based.

**Survival analysis.** Survival analysis was performed using a Cox proportional hazard model with the survival time (censored at 90 days) as outcome and the four longitudinal enzyme activities as independent variables. In addition, a logistic regression model with survival up to day 90 as binary outcome was fitted.

All statistical analyses have been performed using R (Version 3.1.2, R Core Development Team (2008)). A $p$-value below 0.05 was considered significant, unless otherwise stated.

## 3. Results

### 3.1. Study populations

The characteristics of the septic shock patients are summarized in Table 1, including all the inflammatory, hemodynamic and metabolic parameters selected for this study. The septic shock patients were of older age, several of them suffered from multiple conditions, such as various malignancies, hypertension, myocardial infarction, COPD and others. None of the patients, however, used DPP4-inhibitors. The most common sources of sepsis were pneumonia (n = 17), abdominal sepsis (n = 10), mediastinitis (n = 4) and soft tissue/muscular infection (n = 3). In the other patients, the sites of infection were central line sepsis, wound infection, myocarditis and leptospirosis. In 2 patients multiple sites were identified (pneumonia/central line sepsis). The median (IQR) APACHE II score of 25 (20–29), a duration of vasopressor treatment of 3.0 (1.5–8.5) days and the necessity to restart this treatment in 6 out of the 40 patients illustrate the severity of septic shock encountered in these patients. Additionally, 13 patients had died at day 90. More information on the ICU control patients can be found in Table 1 and S1 Table.

### 3.2. Enzyme activities in septic shock patients and ICU controls

The median (IQR) values of the enzymatic activities can be found in Fig 1 and S2 Table. FAP and DPP4 activity were both significantly lower in patients with septic shock compared to ICU controls, across all four days ($p \leq 0.001$). Within the septic shock patients, FAP showed a significant decrease from day 1 to day 3 ($p \leq 0.001$), whereas DPP4 differed between day 1 and days 3 and 5 ($p = 0.003$ and $p = 0.028$). PRCP activity was higher in the septic shock patient group on day 1 ($p = 0.001$) compared to the control patients. Within the septic shock patients, a decrease in PRCP activity was observed between day 1 and day 5 ($p = 0.017$) and day 7 ($p = 0.007$). On all four days of measurement (day 1, 3, 5 and 7), the septic shock patients had a significantly higher PREP activity ($p \leq 0.001$) than the ICU control patients. Some septic shock patients had a very high PREP activity on day 1 but normalized to control levels afterwards. For DPP4, FAP and PREP, the activity levels did not normalize to control levels during the 7-day study period. We also measured the enzyme activities in 30 healthy controls (see S1 Appendix) and compared the values with the ICU controls. Between these two controls groups, no statistically significant differences were identified, and a multivariate analysis via

**Table 1. Study populations: Characteristics and the selected septic shock related parameters.** Longitudinally measured parameters were determined on days 1, 3, 5 and 7 after diagnosis. Data are expressed as median (IQR), unless stated otherwise.

| A. Study populations characteristics | Patients (*n* = 40) | ICU controls (*n* = 22) | | |
|---|---|---|---|---|
| **Age, years (mean ± SD)** | 63 ± 15 | 51 ± 12 | | |
| **Gender** | Female: *n* = 19 | Female: *n* = 13 | | |
| | Male: *n* = 21 | Male: *n* = 9 | | |
| **B. Septic shock parameters in patients [median (IQR)]** | | | | |
| **Longitudinally measured** | **Day 1** | **Day 3** | **Day 5** | **Day 7** |
| **Mean arterial pressure, mmHg** | **75.00** (68.00–80.00) | **76.00** (70.25–83.50) | **76.00** (70.25–87.25) | **82.50** (69.75–98.25) |
| **Heart rate, beats/minute** | **106.00** (90.00–117.75) | **89.50** (80.25–110.25) | **96.50** (77.25–109.75) | **89.00** (75.25–103.50) |
| PaO$_2$/FiO$_2$ ratio | **30.44** (23.40–35.60) | **30.38** (19.77–35.94) | **30.43** (26.89–38.65) | **31.14** (25.46–36.81) |
| Thrombocytes, x 10$^9$/L | **195.00** (141.25–321.50) | **152.00** (90.50–320.25) | **140.00** (65.00–306.00) | **163.50** (97.00–336.00) |
| **Total bilirubin, μmol/L** | **13.00** (9.00–26.00) | **15.00** (8.50–66.75) | **21.50** (12.00–28.25) | **25.50** (8.50–33.75) |
| **Creatinine, μmol/L** | **139.00** (73.75–279.00) | **119.00** (60.75–234.25) | **105.00** (52.00–146.00) | **94.00** (52.00–146.00) |
| Leucocytes, x 10$^9$/L | **14.20** (7.20–19.40) | **14.40** (6.35–19.30) | **11.30** (6.50–16.30) | **11.85** (7.35–17.53) |
| **Lactate, mmol/L** | **1.80** (1.20–2.65) | **1.40** (1.20–1.80) | **1.35** (1.20–2.23) | **1.60** (1.20–1.85) |
| **Noradrenalin infusion rate, μg/kg/min** | **0.20** (0.11–0.50) | **0.10** (0.00–0.30) | **0.00** (0.00–0.10) | **0.00** (0.00–0.08) |
| **TNFα, pg/mL** | **33.42** (19.90–72.00) | **23.63** (15.53–41.46) | **22.90** (14.83–35.42) | **20.93** (11.92–32.90) |
| **IFNγ, pg/mL** | **9.50** (3.20–21.25) | **8.00** (3.20–15.20) | **5.32** (3.20–16.15) | **5.36** (3.20–16.32) |
| **IL-1β, pg/mL** | **3.20** (0.70–3.44) | **3.20** (0.70–3.20) | **3.20** (0.84–3.20) | **3.20** (0.84–3.80) |
| **IL-1RA, pg/mL** | **186.54** (53.10–694.25) | **66.34** (36.93–198.91) | **69.25** (41.65–132.36) | **62.38** (38.54–120.76) |
| **IL-6, pg/mL** | **216.42** (48.18–1489.25) | **45.90** (12.44–137.43) | **25.32** (11.50–76.98) | **31.42** (8.92–61.63) |
| **IL-8, pg/mL** | **135.60** (59.72–327.23) | **54.77** (31.82–141.15) | **45.26** (28.03–110.96) | **50.54** (28.54–99.23) |
| **IL-10, pg/mL** | **78.43** (26.35–262.50) | **38.78** (8.75–96.05) | **27.69** (9.88–54.15) | **26.60** (7.71–44.05) |
| **I-FABP, pg/mL** | **374.00** (192.00–912.90) | **522.13** (187.00–1422.23) | **455.16** (241.09–1324.25) | **789.50** (361.25–1347.32) |
| **Dialysis (yes/total)** | **5/40** | **5/40** | **9/40** | **11/40** |
| **Ventilation (yes/total)** | **36/40** | **32/40** | **32/40** | **29/40** |
| **Measured once** | | | | |
| **Length of hospital stay, days** | **41.00** (28.25–53.75) | | | |
| **Length of Intensive Care stay, days** | **15.50** (9.50–32.50) | | | |
| **APACHE II score** | **25.00** (20.00–29.00) | | | |
| **Duration of noradrenalin treatment, days** | **3.00** (1.50–8.50) | | | |
| **Restart of noradrenalin treatment** | Yes: 6 No: 34 | | | |
| **Mortality at day 90** | 13/40 | | | |
| **SOFA score on day 1 (min-max)** | **8** (4–15) | | | |

Abbreviations used: I-FABP: intestinal fatty-acid binding protein; IFNγ: interferon γ; IL: interleukin; IL-1RA: interleukin-1 receptor antagonist; IQR: interquartile range; PaO$_2$/FiO$_2$ ratio: ratio of arterial oxygen partial pressure to fractional inspired oxygen; TNFα: tumor necrosis factor α.

principal components (see S1 and S2 Figs), gave no indication that the two groups were different.

## 3.3. Receiver operating characteristic curves

Since large differences were found in the activities of all four enzymes, ROC curves were computed to evaluate the ability of each of the enzymes individually to distinguish septic shock patients from ICU controls. DPP4, FAP and PREP turned out to be very good in discriminating septic shock from non-septic shock ICU control patients with areas under the curve (AUC) of 0.86 (95% confidence interval [CI]: 0.77–0.95), 0.94 (95% CI: 0.89–0.99) and 0.88

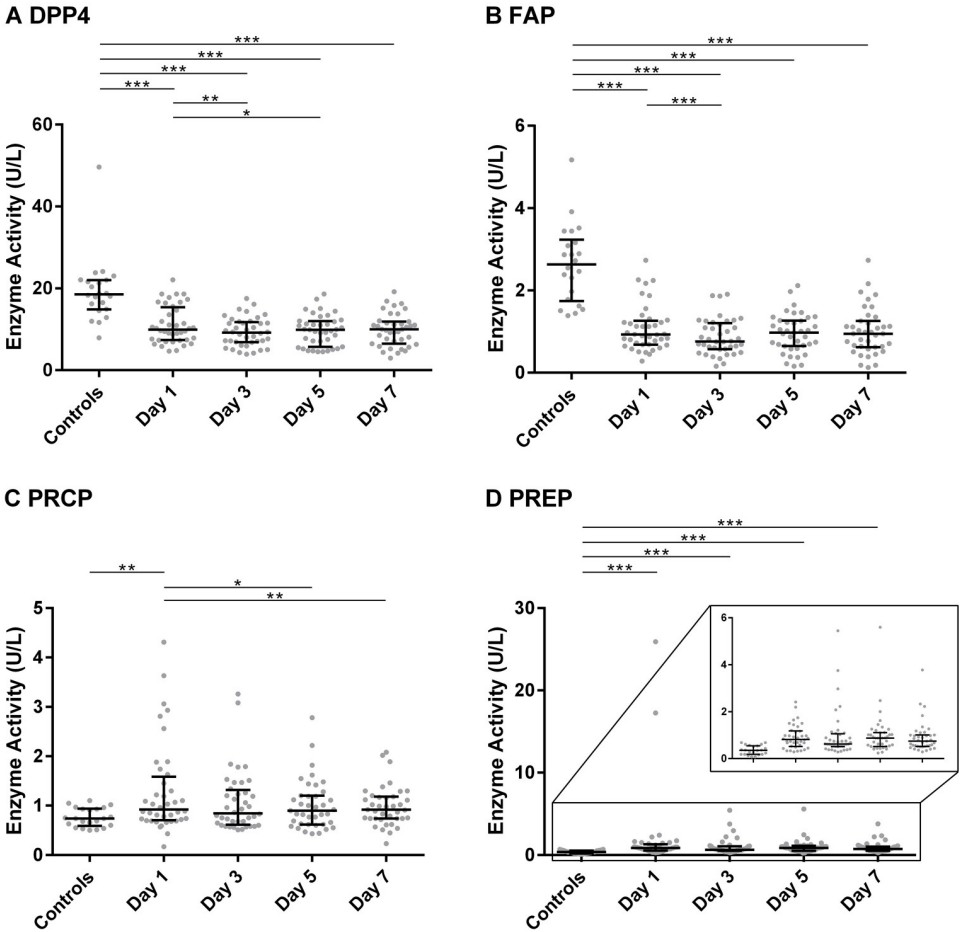

**Fig 1. The plasma activity of DPP4, FAP, PRCP and PREP in septic shock patients and ICU controls.** The plasma activity of DPP4 (A), FAP (B), PRCP (C) and PREP (D), expressed in U/L (median with interquartile range) on days 1, 3, 5 and 7 of the septic shock patients (*n* = 40) and in the ICU control group (*n* = 22). Differences between the control group and the different days in the patients were tested using linear mixed models, followed by a post-hoc analysis with Dunnett's correction. Within the septic shock patients, pairwise differences between the days were modeled using linear mixed models, followed by a post-hoc Tukey's correction. For PREP, the test was carried out on the log-transformed enzyme activity due to the non-normality of the residuals. * p ≤ 0.05; ** p ≤ 0.01; *** p ≤ 0.001. See also S2 Table. Abbreviations used: DPP4: dipeptidyl peptidase 4; FAP: fibroblast activation protein α; PRCP: prolylcarboxypeptidase; PREP: prolyl oligopeptidase; U/L: units per liter.

(95% CI: 0.80–0.96), respectively. PRCP had a lower diagnostic value with an AUC of 0.71 (95% CI: 0.58–0.83). The four ROC curves are depicted in Fig 2. Cutoff values, either determined using the Youden index (giving equal weight to false positives and false negatives) or to obtain 100% sensitivity (to identify all patients with sepsis), are listed in Table 2 together with the sensitivity, specificity, negative and positive predictive value and the negative and positive likelihood ratios. Two maxima were obtained for PRCP, in this case the cutoff that would yield the highest sensitivity was selected, as that would identify more patients with septic shock. The complete dataset can be found in S3 Table (DPP4), S4 Table (FAP), S5 Table (PRCP) and S6 Table (PREP). In some cases, but not always, combining enzymes lead to better predictions. Therefore, we chose to evaluate whether the combination of FAP and DPP4 and FAP with PREP would make a better model. We chose these two combinations since FAP had the highest AUC value and DPP4 and PREP had very similar AUC values. Combining FAP and DPP4

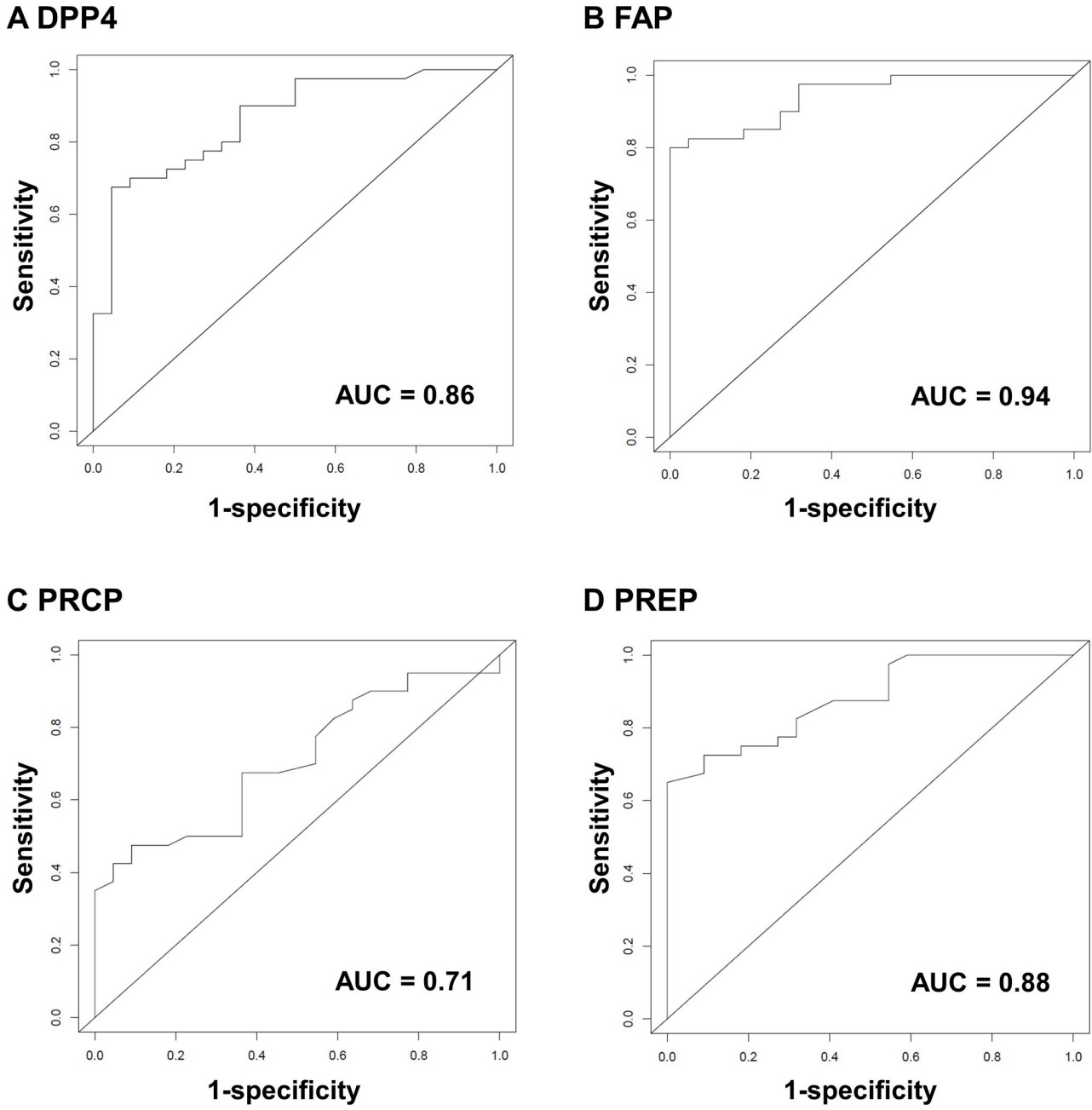

**Fig 2. Receiver operating characteristic (ROC) curves for DPP4, FAP, PRCP and PREP.** ROC curves for DPP4 (A), FAP (B), PRCP (C), PREP (D) measured in EDTA-plasma. Cutoff values are given in Table 2. Also see S3, S4, S5 and S6 Tables. Abbreviations used: AUC: area under the curve; DPP4: dipeptidyl peptidase 4; FAP: fibroblast activation protein α; PRCP: prolylcarboxypeptidase; PREP: prolyl oligopeptidase.

showed an AUC value of 0.94 (95% CI: 0.89–0.99), which is not different from FAP alone. On the other hand, when FAP and PREP were combined as predictors using logistic regression, the AUC value increased to 1 (95% CI: 0.95–1), which would mean that this model correctly predicts at a wide range of cutoffs. However, this latter estimate may be overly optimistic, as the prediction model was never validated in an independent dataset, and the current dataset is too small to carry out cross validation. The ROC curves can be found in S3 Fig.

**Table 2. Cutoff values based upon the ROC curves of DPP4, FAP, PRCP and PREP.** For the septic shock patients day 1 was used. Cutoff values were determined using the Youden index or when all patients with septic shock would be identified (sensitivity = 100%). For DPP4 and FAP, enzymatic activities lower than the indicated cutoff value would suggest a diagnosis with septic shock. For PRCP and PREP this would be the case if the activity lies above the calculated cutoff value. The complete datasets can be found in S3 Table (DPP4), S4 Table (FAP), S5 Table (PRCP) and S6 Table (PREP).

| | DPP4 (Controls: $n$ = 22; Cases: $n$ = 40) | | FAP (Controls: $n$ = 22; Cases: $n$ = 40) | | PRCP (Controls: $n$ = 22; Cases: $n$ = 40) | | PREP (Controls: $n$ = 22; Cases: $n$ = 40)) | |
|---|---|---|---|---|---|---|---|---|
| | Youden index (J = 0.63) | All septic shock patients | Youden index (J = 0.80) | All septic shock patients | Youden index (J = 0.38) | All septic shock patients | Youden index (J = 0.65) | All septic shock patients |
| **Cutoff value, U/L** | 11.41 | 22.06 | 1.31 | 2.73 | 1.03 | 0.17 | 0.69 | 0.29 |
| **Sensitivity** | 0.68 | 1.00 | 0.80 | 1.00 | 0.48 | 1.00 | 0.65 | 1.00 |
| **Specificity** | 0.95 | 0.18 | 1.00 | 0.45 | 0.91 | 0.00 | 1.00 | 0.41 |
| **Positive predictive value** | 0.96 | 0.69 | 1.00 | 0.77 | 0.90 | 0.65 | 1.00 | 0.75 |
| **Negative predictive value** | 0.62 | 1.00 | 0.73 | 1.00 | 0.49 | NA | 0.61 | 1.00 |
| **Positive likelihood ratio** | 14.85 | 1.22 | $\infty$ | 1.83 | 5.23 | 1.00 | $\infty$ | 1.69 |
| **Negative likelihood ratio** | 0.34 | 0.00 | 0.20 | 0.00 | 0.58 | NA | 0.35 | 0.00 |

Abbreviations used: DPP4: dipeptidyl peptidase 4, FAP: fibroblast activation protein α; NA: not applicable; PRCP: prolylcarboxypeptidase; PREP: prolyl oligopeptidase; U/L: units per liter.

### 3.4. Survival analysis

A survival analysis was performed to investigate if there is an association between the enzyme activities and the time till death. At day 90, 13 people had died (32.5%). An overall effect on survival time was found for PRCP and PREP ($p$ = 0.043 and $p$ = 0.048, respectively) in the Cox proportional hazard model. Despite this overall effect, none of the enzymes at the individual time points was significantly associated with mortality before day 90. A nominally significant association was observed between mortality before day 90 and the DPP4 enzymatic activity at day 1 ($p$ = 0.04 and $p$ = 0.03, respectively) in both the Cox proportional hazard model and the logistic regression model, with an increased DPP4 enzymatic activity being associated with an increase in survival.

### 3.5. Association analysis

Because we wanted to see whether the enzymatic activities were associated with other parameters frequently measured in sepsis and septic shock (which are presented in Table 1), we additionally performed an association analysis (see S1 Appendix). Unfortunately, only a few strong associations were identified (PREP with intestinal fatty-acid binding protein (I-FABP) and interleukin-1 receptor antagonist (IL-1RA)). See S4 Fig.

## 4. Discussion

To the best of our knowledge, this is the first report on the activity of the proline-specific peptidases DPP4, PRCP, FAP and PREP in a well-characterized population of septic shock patients over a time-interval of 7 days. Since they are involved in blood pressure regulation and inflammatory pathways, both are overtly disturbed in sepsis and septic shock, it is conceivable that these enzymes are dysregulated or involved in the pathogenesis of sepsis and septic shock. Indeed, both DPP4 and FAP show a significantly lower enzyme activity in plasma of patients with septic shock for all time points. This is in accordance with previously published data reporting a lower DPP4 activity in patients with severe sepsis [9]. Lower plasma DPP4 activity,

however, is not specific for sepsis and has also been reported in other inflammatory disorders such as rheumatoid arthritis [40,41] and in certain cancers as reviewed in [42]. FAP enzyme activity is known to be increased in certain liver diseases and decreased in certain malignancies, but remains unchanged in rheumatoid arthritis or scleroderma [43]. Unfortunately, the exact sources or mechanisms by which these enzymes are released into the bloodstream are not fully understood [44,45]. It is plausible that these secretion or shedding mechanisms are disturbed.

We report that PRCP activity is increased in the septic shock patient group on day 1. As PRCP is an angiotensin II degrading enzyme and angiotensin II is recently approved as therapy for septic shock [27,28], PRCP has the potential to become a promising predictive biomarker for response to angiotensin II. Angiotensin II treatment probably will not have the desired effect in patients with high PRCP activity. A dedicated study, including associations between PRCP activity and angiotensin II levels, angiotensin (1–7) levels and ideally other markers of the renin-angiotensin system is needed to confirm this hypothesis.

At present, there is no good diagnostic biomarker for sepsis (for an overview of biomarkers studied in sepsis we refer to [46,47]). Other biomarkers that have been studied in the past, such as procalcitonin and C-reactive protein, lack specificity [47]. To see whether surgery itself would also influence the enzymatic activities, we chose patients that were admitted to the ICU after major intracranial surgery as a relevant control group. In our study, the enzymatic activity of the ICU controls did not differ from a healthy control group, reassuring that the observed differences between the septic shock patients and ICU control patients are indeed due to the disease state of the septic shock patients and not to the surgery or the underlying disease of the ICU control patients. Additionally, reference values for DPP4 have been published in the past, and further confirm that the observed DPP4 enzymatic activities are lower in the septic shock patients, while the ICU control group falls within the expected range [48].

DPP4, FAP and PREP can discriminate between the septic shock patients (day 1) and the ICU control group, as reflected by high AUC values. When the Youden index is applied, FAP reaches the highest sensitivity (80%) and specificity (100%). Since septic shock has a high mortality, another cutoff value was calculated in which no patients would be missed (sensitivity is 100%). In this case, each of the enzymes gives an insufficient specificity as more than half of the ICU control patients would be falsely diagnosed with septic shock and would receive unnecessary treatment. The ability of the enzymes to discriminate between an ICU control group and septic shock patients, does not necessarily mean that these enzymes are indeed good biomarkers for sepsis itself. Septic shock is a very extreme condition, which does not need a specific biomarker. A biomarker for sepsis is needed in the early stage, when one is not sure whether a patient does indeed have sepsis or not. Therefore, in the future, it would be very interesting to study larger groups of patients and to prospectively include all patients at high risk of developing sepsis and measure, for example, their DPP4- and FAP-activities on a daily basis. This would provide us with the information whether the observed reduction in enzymatic activity in the most extreme form of sepsis develops gradually or at once and whether DPP4 and/or FAP should be used as a target to watch in patients with a high risk of developing sepsis. Thereby, identifying these patients earlier and enabling the caregivers to initiate therapy earlier on, in order to improve the outcome of these patients. Most likely, a combination of these enzymes with other parameters used in sepsis could result in a diagnostic biomarker panel. For now, especially FAP looks very promising and also the combination of FAP and PREP should be further explored. A limitation of this study is that only samples from patients who stayed in the ICU during the entire week, were available. Follow-up studies should include all patients diagnosed with sepsis. This may even increase the biomarker value of the enzymes studied here. Moreover, a good biomarker should be cheap and easily

measurable, for DPP4 this is certainly the case, since it only requires a commercially available, chromogenic substrate and results can be obtained within 1 hour after blood sampling. We recently published a new and faster method for the simultaneous measurement of FAP and PREP using a commercially available fluorogenic substrate, making also these measurements readily available [37].

Only a marginally significant association with mortality at day 90 was found for DPP4 activity on day 1. An overall association with PREP and PRCP was seen in the Cox proportional hazard model. However, given the large number of associations tested here, the reported p-values are not very promising and therefore we cannot conclude from this data that PRCP or PREP are good prognostic biomarkers for sepsis. Additionally, it should be taken into consideration that only patients, of whom samples on days 1, 3, 5 and 7 were available, were included in the study. A stronger association might be identified when the DPP4-activity on day 1 is measured in a larger and more diverse sepsis patient population.

In the association analysis, a large panel of parameters was studied. The selection of the parameters was based upon their use as biomarkers in sepsis, their involvement in inflammation and/or importance for hemodynamic patient characterization. Rather surprisingly, only a few strong associations were found. Most remarkable are the associations between PREP and I-FABP and PREP with IL-1RA. I-FABP is quickly released into the circulation after intestinal damage [49]. Since PREP is also highly expressed in the gastro-intestinal tract [50,51] and intestinal damage occurs in sepsis [52], it is possible that PREP could serve as a blood-based biomarker for intestinal damage. The associations between PREP and IL-1RA and PREP and I-FABP are new and warrant further research.

## 5. Conclusion

We show here that the activities of the proline-specific peptidases DPP4, PRCP, FAP and PREP are dysregulated in septic shock patients and can significantly discriminate between septic shock patients and the selected ICU control group (PREP, FAP and DPP4: AUC values of 0.88 (CI: 0.80–0.96), 0.94 (CI: 0.89–0.99) and 0.86 (CI: 0.77–0.95)).

The present study strongly supports a further exploration of these enzymes as potential diagnostic and/or prognostic biomarkers in sepsis and septic shock.

## Supporting information

**S1 Fig. Plasma activity of DPP4, FAP, PRCP and PREP in healthy and ICU controls.** The plasma activity of DPP4 (A), FAP (B), PRCP (C) and PREP (D), expressed in U/L (median with interquartile range), measured in a healthy control group (n = 29–30) and the ICU control group (n = 22). Differences between the two control groups were tested using Mann-Whitney U tests. Abbreviations used: DPP4: dipeptidyl peptidase 4; FAP: fibroblast activation protein α; NS: non-significant; PRCP: prolylcarboxypeptidase; PREP: prolyl oligopeptidase; U/L: units per liter.
(DOCX)

**S2 Fig. Principal component analysis.** Plot of the first two principal components scores for the ICU control group and the healthy control group, based upon PREP, PRCP, FAP and DPP4. Abbreviations used: ICU; intensive care unit; PC: principal component.
(DOCX)

**S3 Fig. Receiver Operating Characteristic (ROC) curves of the combination FAP and DPP4 and FAP with PREP.**
(DOCX)

**S4 Fig. Visual representation of the statistical analysis of the studied associations.** (A) Dot plot of the log-transformed $p$-value for the interaction ($p$Int) for the longitudinally measured parameters. The dashed line indicates a $p$-value of 0.01. Significant interactions were obtained for PRCP with total bilirubin and TNFα. P-values are based upon linear mixed models. (B) Dot plot of the log-transformed $p$-value for the main effect ($p$Main) for the longitudinally measured parameters. In case of a significant interaction between time and the parameter, no main effect $p$-value is shown in the dot plot. The dashed line indicates a $p$-value of 0.01. Linear mixed models were fitted to obtain the $p$-values. (C) $R^2$ for the longitudinally measured parameters. $R^2$ is only depicted when $p$Int > 0.01. P-values are based upon linear mixed models. ∘ DPP4; + PRCP; △ FAP; ● PREP. Abbreviations used: DPP4: dipeptidyl peptidase 4; FAP: fibroblast activation protein α; I-FABP: intestinal fatty acid-binding protein; IFNγ: interferon γ; IL: interleukin; IL-1RA: interleukin-1 receptor antagonist; MAP: mean arterial pressure; NIR, noradrenalin infusion rate; $PaO_2/FiO_2$ ratio: ratio of arterial oxygen partial pressure to fractional inspired oxygen; PRCP: prolylcarboxypeptidase; PREP: prolyl oligopeptidase; TNFα: tumor necrosis factor α.
(DOCX)

**S1 Table. Characteristics of the non-septic shock ICU control group.** A total of 22 patients were included for this study.
(DOCX)

**S2 Table. The enzyme activity of DPP4, FAP, PRCP and PREP.** Enzymatic activities were determined in EDTA-plasma of septic shock patients on day 1, 3, 5 and 7 and non-septic shock ICU control patients. Data are represented as median (interquartile range). Abbreviations used: DPP4: dipeptidyl peptidase 4; FAP: fibroblast activation protein α; PRCP: prolyl-carboxypeptidase; PREP: prolyl oligopeptidase; U/L: units per liter.
(DOCX)

**S3 Table. Cutoff values receiver operating characteristic curve of dipeptidyl peptidase 4 (DPP4).** The 1-specificity, sensitivity, negative predictive value (NPV), positive predictive value (PPV), positive likelihood ratio (LR+), negative likelihood ratio (LR-) and Youden index for every cutoff value expressed in U/L for DPP4. For the septic shock patients day 1 was used. Blue indicates the maximum for the Youden index. Red indicates where the sensitivity reaches 1. In the case of DPP4, a value lower than the indicated cutoff would suggest a diagnosis with septic shock. ICU controls: n = 22; Septic shock patients: n = 40.
(DOCX)

**S4 Table. Cutoff values receiver operating characteristic curve of fibroblast activation protein α (FAP).** The 1-specificity, sensitivity, negative predictive value (NPV), positive predictive value (PPV), positive likelihood ratio (LR+), negative likelihood ratio (LR-) and Youden index for every cutoff value expressed in U/L for FAP. For the septic shock patients day 1 was used. Blue indicates the maximum for the Youden index. Red indicates where the sensitivity reaches 1. In the case of FAP, a value lower than the indicated cutoff would suggest a diagnosis with septic shock. ICU controls: n = 22; Septic shock patients: n = 40.
(DOCX)

**S5 Table. Cutoff values receiver operating characteristic curve of prolylcarboxypeptidase (PRCP).** The 1-specificity, sensitivity, negative predictive value (NPV), positive predictive value (PPV), positive likelihood ratio (LR+), negative likelihood ratio (LR-) and Youden index for every cutoff value expressed in U/L for PRCP. For the septic shock patients day 1 was used. Blue indicates the maximum for the Youden index. Two maxima were obtained for PRCP, in

this case the cutoff that would yield the highest sensitivity was selected, as that would identify more patients with septic shock. Red indicates where the sensitivity reaches 1. In the case of PRCP, a value higher than the indicated cutoff would suggest a diagnosis with septic shock. ICU controls: n = 22; Septic shock patients: n = 40.
(DOCX)

**S6 Table. Cutoff values receiver operating characteristic curve of prolyl oligopeptidase (PREP).** The 1-specificity, sensitivity, negative predictive value (NPV), positive predictive value (PPV), positive likelihood ratio (LR+), negative likelihood ratio (LR-) and Youden index for every cutoff value expressed in U/L for PREP. For the septic shock patients day 1 was used. Blue indicates the maximum for the Youden index. Red indicates where the sensitivity reaches 1. In the case of PREP, a value higher than the indicated cutoff would suggest a diagnosis with septic shock. ICU controls: n = 22; Septic shock patients: n = 40.
(DOCX)

**S1 Appendix. Dysregulated activities of proline-specific enzymes in septic shock patients (sepsis-2).**
(DOCX)

## Author Contributions

**Conceptualization:** Gwendolyn Vliegen, Kaat Kehoe, Esther Peters, Peter Pickkers, Philippe G. Jorens, Ingrid De Meester.

**Data curation:** Gwendolyn Vliegen, Bart 's Jongers, Esther Peters.

**Formal analysis:** Gwendolyn Vliegen, Kaat Kehoe, An Bracke, Emilie De Hert, Robert Verkerk, Erik Fransen, Esther Peters.

**Funding acquisition:** Anne-Marie Lambeir, Samir Kumar-Singh, Ingrid De Meester.

**Investigation:** Gwendolyn Vliegen, Kaat Kehoe, An Bracke, Emilie De Hert, Robert Verkerk, Esther Peters.

**Methodology:** Gwendolyn Vliegen, Kaat Kehoe, Erik Fransen, Bart 's Jongers, Esther Peters, Anne-Marie Lambeir, Peter Pickkers, Ingrid De Meester.

**Project administration:** Gwendolyn Vliegen, Bart 's Jongers, Esther Peters, Philippe G. Jorens, Ingrid De Meester.

**Resources:** Bart 's Jongers, Esther Peters, Anne-Marie Lambeir, Samir Kumar-Singh, Peter Pickkers, Ingrid De Meester.

**Supervision:** Anne-Marie Lambeir, Samir Kumar-Singh, Peter Pickkers, Philippe G. Jorens, Ingrid De Meester.

**Visualization:** Gwendolyn Vliegen.

**Writing – original draft:** Gwendolyn Vliegen, Bart 's Jongers, Esther Peters.

**Writing – review & editing:** Kaat Kehoe, An Bracke, Emilie De Hert, Robert Verkerk, Erik Fransen, Anne-Marie Lambeir, Samir Kumar-Singh, Peter Pickkers, Philippe G. Jorens, Ingrid De Meester.

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
