## [Decision Letter · Decision Letter 0]

12 Dec 2019

PONE-D-19-30816

Dysregulated activities of proline-specific enzymes in septic shock patients (sepsis-2)

PLOS ONE

Dear Full professor De Meester,

Thank you for submitting your manuscript to PLOS ONE. After careful consideration, we feel that it has merit but does not fully meet PLOS ONE’s publication criteria as it currently stands. Therefore, we invite you to submit a revised version of the manuscript that addresses the points raised during the review process.

Three external reviewers each gave specific comments on your manuscript (see details below). Their main questions are related to additional information on the clinical parameters/history of the patient and control samples and further analysis of potential correlations. Provide an answer to their comments in a point-by-point reply. Also clearly indicate in the manuscript that the patient population is limited if no additional samples are available. The legends of the figures need to be included in the text according to the Journal's recommendations. These will be combined with the correct figures once the manuscript would be accepted for publication.

We would appreciate receiving your revised manuscript by Jan 26 2020 11:59PM. To enhance the reproducibility of your results, we recommend that if applicable you deposit your laboratory protocols in protocols.io, where a protocol can be assigned its own identifier (DOI) such that it can be cited independently in the future. For instructions see: http://journals.plos.org/plosone/s/submission-guidelines#loc-laboratory-protocols

We look forward to receiving your revised manuscript.

Kind regards,

Paul Proost, Ph.D.

Academic Editor

PLOS ONE

Journal Requirements:

Reviewers' comments:

Reviewer's Responses to Questions

**Comments to the Author**

1. Is the manuscript technically sound, and do the data support the conclusions?

Reviewer #1: Yes

Reviewer #2: Partly

Reviewer #3: Partly

2. Has the statistical analysis been performed appropriately and rigorously? 

Reviewer #1: Yes

Reviewer #2: No

Reviewer #3: Yes

3. Have the authors made all data underlying the findings in their manuscript fully available?

Reviewer #1: Yes

Reviewer #2: No

Reviewer #3: Yes

4. Is the manuscript presented in an intelligible fashion and written in standard English?

Reviewer #1: Yes

Reviewer #2: Yes

Reviewer #3: Yes

5. Review Comments to the Author

Reviewer #1: This paper is very interestiong and well written. But I have several concerns.

Major points.

1)What is the cause of sepsis? Pneumonia? peritonitis? Trauma? Tumor? You should show the cause of sepsis.

2)Please show the SOFA score (Day 1,3,5,7) in the sepsis group.

3)Were there past medical history, especially diabetis? Was there a patient taking a DPP4 inhibitor? If so, did you think that DPP4 inhibitor have an effect on this outcome?

4) Were there case where steroids were used for refractory shock? If so, did you think that steroid administration have an effect on this outcome?

Reviewer #2: PLOS ONE – VLIEGEN ET AL – DECEMBER 2019

SUMMARY

These authors investigated the activity of several proline-specific enzymes (PSE) (dipeptidyl peptidase 4 (DPP4), prolylcarboxypeptidase (PRCP), fibroblast activation protein α (FAP) and prolyl oligopeptidase (PREP)) in 40 septic shock patients, 22 ICU controls (neurosurgical patients) and 30 healthy controls. PRCP and PREP were higher while FAP and DPP4 activity were lower in septic shock than in controls (as per Figure 1). There were AUCs of 0.86 – 0.94 with septic shock. The PSE activity levels of the healthy and ICU controls dd not differ. These PSE cleave proteins near proline residues. The investigators used their own lab-developed activity assays. The study is novel and provides further insights into the roles of PSE in septic shock. It is a small study so only really opens the door to the role of these enzymes as septic shock biomarkers.

MAJOR

1. Results: “Combination of FAP and DPP4 did not increase the AUC value compared to FAP alone.” This is a random statement because combinations were not mentioned in methods and no other combinations are mentioned. Please delete or do a thorough combination analysis. Combinations of biomarkers are often better, and I would recommend taking this on as an additional analysis. Indeed, the authors make this statement themselves in the discussion: “Most likely, a combination of these enzymes with other parameters used in sepsis could result in a diagnostic biomarker panel.”

2. It would be useful to understand further how these enzyme activities are associated with altered clinical outcomes in addition to mortality. I would expect and like to see whether enzyme activity was associated with organ dysfunction as assessed by days alive and free of vasopressors, ventilation and renal replacement therapy (RRT). The enzyme activity is associated with mortality, but we don’t know if that also alters cardiovascular, respiratory or renal function as reflected by the days alive and free of vasopressors, ventilation and renal replacement therapy (RRT).

3. The sample size is relatively small and there no validation cohort. This is a critical limitation of believing the credibility of the results herein.

4. The statistical analysis for the diagnostic aim should examine in addition only the day 1 values, since there is survival/non-survivor risk of bias due to early deaths after day 1. There could be non-random drop out due to death that could bias the results herein. The authors state in the discussion:” Additionally, it should be taken into consideration that only patients, of whom samples on days 1, 3, 5 and 7 were available, were included in the study.”

5. PRCP modifies/ presumably lowers angiotensin II levels (as per Yang HYT, Erdös EG, Chiang TS. New Enzymatic Route for the Inactivation of. Angiotensin. Nature. 1968;218(5148):1224–6) and angiotensin II is a recently approved therapy for septic shock(1, 2) so that has greater relevance for possibly discovering a predictive biomarker for response to angiotensin II. PRCP levels were increased in septic shock and higher levels were associated with a poor prognosis in your study. Furthermore, there are good studies of abnormalities of the renal-angiotensin-aldosterone (RAAS) system in sepsis(3). Accordingly, it would be useful to examine the PRCP association with markers of RAAS actively (plasma rein activity etc.). Of note, PRCP was not a great diagnostic (Low AUC) but was a good biomarker of prognosis – this suggests that the RAAS is a better marker of recovery or not from septic shock. Also, pre-sepsis use of ACE inhibitors could modify the PRCP activity – do you have that information available?

MINOR

1. Results. Study populations paragraph: would be useful to state the mortality rates in the Table 1 and that paragraph.

2. 30 healthy controls aren’t mentioned until the results section. These should be introduced and described in methods (re how and where they were recruited). They should also be described re age, gender etc. in Table 1.

3. The figure legends appear in the text but should be moved to the end.

4. Results, page 15. “For the septic shock patients day 1 was used.” I presume this refers to the ROC for diagnostic biomarker. Please clarify and introduce the rationale for this in methods.

5. Discussion:” Since septic shock has a high mortality, another cutoff value was calculated in which no patients would be missed. In this case, each of the enzymes gives an insufficient specificity as more than half of the ICU control patients would be falsely diagnosed with septic shock.” This is not clear and requires better explanation.

6. The Kaplan-Meier 90-day survival curves for the best PSE activity cutoffs should be shown as a figure in the main manuscript. The reader would like to see visually how PSE levels are associated with prognosis in septic shock.

REFERENCES

1. Khanna A, English SW, Wang XS, Ham K, Tumlin J, Szerlip H, et al. Angiotensin II for the Treatment of Vasodilatory Shock. N Engl J Med. 2017;377(5):419-30.

2. Senatore F, Jagadeesh G, Rose M, Pillai VC, Hariharan S, Liu Q, et al. FDA Approval of Angiotensin II for the Treatment of Hypotension in Adults with Distributive Shock. Am J Cardiovasc Drugs. 2019;19(1):11-20.

3. Emdin M, Fatini C, Mirizzi G, Poletti R, Borrelli C, Prontera C, et al. Biomarkers of activation of renin-angiotensin-aldosterone system in heart failure: how useful, how feasible? Clin Chim Acta. 2015;443:85-93.

Reviewer #3: The manuscript highlights markers that have the potential to become biomarkers for sepsis, for which currently no concrete association exists (as mentioned by authors as well). However, the associations described for most of the proline-specific enzymes were not very strong. As the authors mentioned further research is warranted. This manuscript will greatly benefit from more detailed analysis of age, gender, ethnicity to determine where the enzymes have any stronger correlation in any specific group. It is understandable data from human patients are not easily available however, to correlate the enzymes with progression in sepsis, this study requires control known inflammatory/anti-inflammatory patterns demonstrated side by side. This analysis will reveal the correlation of the inflammatory status, disease progression of sepsis 2 patient along with the enzymes can be potential biomarkers.

6. PLOS authors have the option to publish the peer review history of their article (what does this mean?). If published, this will include your full peer review and any attached files.

Reviewer #1: No

Reviewer #2: No

Reviewer #3: No

---

## [Author Response · Author response to Decision Letter 0]

28 Feb 2020

General: 

1.) Also clearly indicate in the manuscript that the patient population is limited if no additional samples are available. 

Next to what was already present in the text (lines 428-430) we added the following sentence to the Discussion: “A limitation of this study is that only samples from patients who stayed in the ICU during the entire week, were available.” (lines 414-415)

2.) The legends of the figures need to be included in the text according to the Journal's recommendations. These will be combined with the correct figures once the manuscript would be accepted for publication.

We provide the figure legends in the manuscript where the figure should be included, as is stated in the submission guidelines.

3.) Please ensure that your manuscript meets PLOS ONE's style requirements, including those for file naming.

We have split the supporting information into S1 Appendix, S1 Fig, S2 Fig, S3 Fig, S4 Fig, S1 Table, S2 Table, S3 Table, S4 Table, S5 Table and S6 Table. We apologize for the mistake and hope that we now meet PLOS ONE’s style requirements.

Reviewer #1: 

We would like to thank the reviewer for the careful revision of the manuscript and for the useful suggestions. We will address all remarks in detail below.

Major points:

1.) What is the cause of sepsis? Pneumonia? peritonitis? Trauma? Tumor? You should show the cause of sepsis.

We have added this information into 3.1 Study populations (lines 224-228). 

“The most common sources of sepsis were pneumonia (n = 17), abdominal sepsis (n = 10), mediastinitis (n = 4) and soft tissue/muscular infection (n = 3). In the other patients, the sites of infection were central line sepsis, wound infection, myocarditis and leptospirosis. In 2 patients multiple sites were identified (pneumonia/central line sepsis).”

2.) Please show the SOFA score (Day 1,3,5,7) in the sepsis group.

The SOFA score has been assessed on day 1. This information has been included in Table 1 and was analysed statistically (see S1 appendix lines 39-41), to see whether there were associations with the enzymatic activities on day 1. However, none of the enzymatic activities was significantly associated with the SOFA score on day 1 (data not shown).

3.) Were there past medical history, especially diabetis? Was there a patient taking a DPP4 inhibitor? If so, did you think that DPP4 inhibitor have an effect on this outcome?

The patients studied herein, were of older age, several of them suffered from multiple conditions, such as various malignancies, hypertension, myocardial infarction, COPD and others. A sentence describing these patients has been added to 3.1 Study populations (lines 221-223).

“The septic shock patients were of older age, several of them suffered from multiple conditions, such as various malignancies, hypertension, myocardial infarction, COPD and others.”

Of the 40 patients, 6 patients had diabetes. As mentioned in the Introduction “A nested case-control study in type 2 diabetes patients admitted for sepsis did not find a significant association between the use of a DPP4-inhibitor and the development of sepsis [1].” None of the patients used a DPP4-inhibitor. This information has also been added to 3.1 Study populations (line 223-224).

4.) Were there case where steroids were used for refractory shock? If so, did you think that steroid administration have an effect on this outcome?

Multiple patients indeed received corticosteroids during their course. Our study has not been designed to evaluate the effect of corticosteroid use on the enzymatic activities, but limited data on this subject can be found in the literature for DPP4 and we do not expect an effect of (acute) corticosteroid treatment on serum DPP4 enzymatic activity [2,3]. On the other enzymes, we could not find literature data and it would be an interesting idea for the future to look into the effect of corticosteroid treatment on the enzymatic activity.

Reviewer #2: 

First of all, we would like to thank the reviewer for the useful suggestions. We hope that our answers and adjustments are within the expectations.

SUMMARY

These authors investigated the activity of several proline-specific enzymes (PSE) (dipeptidyl peptidase 4 (DPP4), prolylcarboxypeptidase (PRCP), fibroblast activation protein α (FAP) and prolyl oligopeptidase (PREP)) in 40 septic shock patients, 22 ICU controls (neurosurgical patients) and 30 healthy controls. PRCP and PREP were higher while FAP and DPP4 activity were lower in septic shock than in controls (as per Figure 1). There were AUCs of 0.86 – 0.94 with septic shock. The PSE activity levels of the healthy and ICU controls dd not differ. These PSE cleave proteins near proline residues. The investigators used their own lab-developed activity assays. The study is novel and provides further insights into the roles of PSE in septic shock. It is a small study so only really opens the door to the role of these enzymes as septic shock biomarkers.

MAJOR

1.) Results: “Combination of FAP and DPP4 did not increase the AUC value compared to FAP alone.” This is a random statement because combinations were not mentioned in methods and no other combinations are mentioned. Please delete or do a thorough combination analysis. Combinations of biomarkers are often better, and I would recommend taking this on as an additional analysis. Indeed, the authors make this statement themselves in the discussion: “Most likely, a combination of these enzymes with other parameters used in sepsis could result in a diagnostic biomarker panel.”

We fully agree with the reviewer that we should clarify how the combination analysis was done. Therefore, we have added the following sentence to 2.4. Statistical analysis (lines 207-210):

“To assess combinations of enzymes as predictors of septic shock, the two enzymes were entered as independent variables into a logistic regression model with disease status as dependent variable. Subsequently, the predicted probability of this model was used as an input for the ROC-curve upon which the AUC calculation was based.”

The sentence “Combination of FAP and DPP4 did not increase the AUC value compared to FAP alone.” in the results section has been replaced with (lines 285-295):

“In some cases, but not always, combining enzymes lead to better predictions. Therefore, we chose to evaluate whether the combination of FAP and DPP4 and FAP with PREP would make a better model. We chose these two combinations since FAP had the highest AUC value and DPP4 and PREP had very similar AUC values. Combining FAP and DPP4 showed an AUC value of 0.94 (95% CI : 0.89 – 0.99) , which is not different from FAP alone. On the other hand, when FAP and PREP were combined as predictors using logistic regression, the AUC value increased to 1 (95% CI : 0.95-1), which would mean that this model correctly predicts at a wide range of cutoffs. However, this latter estimate may be overly optimistic, as the prediction model was never validated in an independent dataset, and the current dataset is too small to carry out cross validation. The ROC curves can be found in S3 Fig.”

In the Discussion the text has been adapted to (lines 411-417):

“Most likely, a combination of these enzymes with other parameters used in sepsis could result in a diagnostic biomarker panel. For now, especially FAP looks very promising and also the combination of FAP and PREP should be further explored. A limitation of this study is that only samples from patients who stayed in the ICU during the entire week, were available. Follow-up studies should include all patients diagnosed with sepsis. This may even increase the biomarker value of the enzymes studied here.”

2.) It would be useful to understand further how these enzyme activities are associated with altered clinical outcomes in addition to mortality. I would expect and like to see whether enzyme activity was associated with organ dysfunction as assessed by days alive and free of vasopressors, ventilation and renal replacement therapy (RRT). The enzyme activity is associated with mortality, but we don’t know if that also alters cardiovascular, respiratory or renal function as reflected by the days alive and free of vasopressors, ventilation and renal replacement therapy (RRT).

We assessed the mortality using a Cox proportional hazard model with the survival time (censored at 90 days) as outcome and the four longitudinal enzyme activities as independent variables. In addition, a logistic regression model with survival up to day 90 as binary outcome was fitted. The first model takes the days alive into account, while the second model only looks at whether the patient is still alive at day 90, yes or no. We describe these two models in the Materials and Methods, 2.4 Statistical analysis and the results thereof in lines 340-348 (only for DPP4 a significant association was seen at day 1 in both models). We hope that this is an answer to the first question.

In our association analysis we included a multitude of parameters, unfortunately we did not identify any strong associations and we therefore decided to describe the statistical analysis and the results in the supporting information. For clarity, we decided for the parameters measured once to only explicitly mention those parameters for which significance was reached (here defined as a p-value below 0.01, due to the multitude of analyses performed). Therefore, only the association between DPP4 and the length of hospital stay is discussed in 2.3. in the supporting information. The p-values of the parameters measured longitudinally are visually represented in S4 Fig.

Regarding vasopressor treatment, we evaluated the effect of the enzymatic activity on the duration of vasopressor treatment and whether this needed to be restarted. However, none of the enzymes had a significant effect on these parameters and are therefore not mentioned in the Results section of the supplementary materials.

We included the effect of the enzymes on the PaO2/FiO2 ratio, but again, no significance was reached here (see S4 Fig).

The data on whether the patients required dialysis or ventilation were available and were therefore included into the manuscript. See Table 1 and the supporting information S1 Appendix. In 1.3. Statistical analysis we have added (line 38) :

 “The parameters ventilation and dialysis were analyzed using generalized linear mixed models.” and in 2.2. the following sentence was added (lines 86-87):

“Additionally, we fitted generalized linear mixed models to test if the enzymes were associated with the necessity of dialysis or ventilation, but no significant associations were found.”

In the main document a reference to the S1 Appendix has been added in line 353.

3.) The sample size is relatively small and there no validation cohort. This is a critical limitation of believing the credibility of the results herein.

We agree with the reviewer that our sample size is rather small. However, as also mentioned by the reviewer in the summary above, it only opens the door for these enzymes as biomarkers in septic shock. We believe that the data presented here is already of significant importance to be distributed within the scientific community, as these enzymes have never been studied before in this setting. We hope that it can be used as a kick-start for further research, including larger studies.

4.) The statistical analysis for the diagnostic aim should examine in addition only the day 1 values, since there is survival/non-survivor risk of bias due to early deaths after day 1. There could be non-random drop out due to death that could bias the results herein. The authors state in the discussion:” Additionally, it should be taken into consideration that only patients, of whom samples on days 1, 3, 5 and 7 were available, were included in the study.”

We fully agree with the reviewer and we have indeed mentioned this in the Discussion. Unfortunately, we do not have the samples of other patients than the 40 samples studied here. We acknowledge that this is a limitation of our study and we will explicitly mention this in the text (lines 414-417).

“A limitation of this study is that only samples from patients who stayed in the ICU during the entire week, were available. Follow-up studies should include all patients diagnosed with sepsis. This may even increase the biomarker value of the enzymes studied here.”

However, we believe that our results should already be disseminated within the scientific community, as we stated above.

5.) PRCP modifies/ presumably lowers angiotensin II levels (as per Yang HYT, Erdös EG, Chiang TS. New Enzymatic Route for the Inactivation of. Angiotensin. Nature. 1968;218(5148):1224–6) and angiotensin II is a recently approved therapy for septic shock(1, 2) so that has greater relevance for possibly discovering a predictive biomarker for response to angiotensin II. PRCP levels were increased in septic shock and higher levels were associated with a poor prognosis in your study. Furthermore, there are good studies of abnormalities of the renal-angiotensin-aldosterone (RAAS) system in sepsis(3). Accordingly, it would be useful to examine the PRCP association with markers of RAAS actively (plasma rein activity etc.). Of note, PRCP was not a great diagnostic (Low AUC) but was a good biomarker of prognosis – this suggests that the RAAS is a better marker of recovery or not from septic shock. Also, pre-sepsis use of ACE inhibitors could modify the PRCP activity – do you have that information available?

We agree with the reviewer that PRCP can be a promising predictive biomarker for response to angiotensin II treatment, as high PRCP levels (seen in septic shock patients) can theoretically reduce circulating Ang II levels. Therefore, more developed research is necessary to investigate whether high circulating PRCP levels have an influence on Ang II levels and administrated Ang II. In a recent study in mice, it was seen that after administration of Ang II, the Ang II conversion to Ang 1-7 in the circulation was independent of PRCP and ACE2, but dependent of PREP (Serfozo et al. Hypertension: 2020;75:173–182). As the reviewer suggested, it is therefore of great relevance to examine the PRCP association with markers of the renin-angiotensin system and definitely with Ang II and Ang 1-7 levels. In this study, we do not have these data available and there are no longer samples available to measure these parameters. We thank the reviewer for this interesting feedback and suggestion, but research into predictive biomarkers falls outside the scope of this study. However, we included an extra paragraph in the discussion to state this interesting hypothesis (lines 374-380):

“We report that PRCP activity is increased in the septic shock patient group on day 1. As PRCP is an angiotensin II degrading enzyme and angiotensin II is recently approved as therapy for septic shock [27,28], PRCP has the potential to become a promising predictive biomarker for response to angiotensin II. Angiotensin II treatment probably will not have the desired effect in patients with high PRCP activity. A dedicated study, including associations between PRCP activity and angiotensin II levels, angiotensin (1-7) levels and ideally other markers of the renin-angiotensin system is needed to confirm this hypothesis.”

Also, in the introduction we mention that Ang II is recently approved as therapy for septic shock, including the references suggested by the reviewer (Senatore, 2019 and Khanna, 2017) (lines 98-101):

“PRCP is involved in the regulation of blood pressure and hypotension is common in sepsis and septic shock, moreover, angiotensin II is recently approved as therapy for distributive shock. Therefore, a possible role for PRCP in the pathogenesis of sepsis and septic shock is conceivable.”

The overall association between PRCP and survival time (p=0.043), formally tests the null hypothesis that all 4 regression coefficients (for day 1, day 3, day 5 and day 7) from the Cox Proportional Hazard model are simultaneously zero. However, none of the regression coefficients from the separate days is significant, and there is no clear trend in the effect of PRCP on survival when going from day 1 to day 7. Therefore, the effect of PRCP on survival remains unclear at the moment. Due to the multitude of hypotheses tested in this study, it cannot be excluded that this association represents a false positive finding.

We stated in the text: “An overall effect on survival time was found for PRCP and PREP (p = 0.043 and p = 0.048, respectively) in the Cox proportional hazard model. Despite this overall effect, none of the enzymes at the individual time points was significantly associated with mortality before day 90”: “An overall association with PREP and PRCP was seen in the Cox proportional hazard model. However, finding such p-values is not surprising given the large number of p-values tested here.” 

To be more clear, we adapted the text to (lines 425-428):

“An overall association with PREP and PRCP was seen in the Cox proportional hazard model. However, given the large number of associations tested here, the reported p-values are not very promising and therefore we cannot conclude from this data that PRCP or PREP are good prognostic biomarkers for sepsis.”

We do not have data available on the pre-sepsis use of ACE-inhibitors. Therefore, we cannot investigate whether the use of ACE inhibitors has an effect on the PRCP activity. Searching the literature, we cannot find information regarding the effects of ACE-inhibitors on PRCP activity, except that a polymorphism (E112D) in the PRCP gene has been associated with blood pressure response to benazepril (Zhang et al. Chin Med J 2009;122:2461-5). Because both, ACE and PRCP, are involved in the (alternative) renin-angiotensin system, ACE inhibitors can possibly have an effect on the PRCP activity, therefore more research is needed which falls outside the scope of this study.

MINOR

1.) Results. Study populations paragraph: would be useful to state the mortality rates in the Table 1 and that paragraph.

We have added a sentence describing the mortality at day 90 to the Study populations (line 231).

“Additionally, 13 patients had died at day 90.”

This information has also been included in Table 1.

We kept the sentence “At day 90, 13 people had died (32.5%).” in the Survival analysis section, to remind the reader.

2.) 30 healthy controls aren’t mentioned until the results section. These should be introduced and described in methods (re how and where they were recruited). They should also be described re age, gender etc. in Table 1.

To avoid confusion with the ICU control group, we decided to provide the information on the healthy controls in the supporting information. We acknowledge that they should be mentioned earlier in the main text. Therefore, we have added a sentence to the Materials and Methods, referring the reader earlier to the supporting information (lines 135-137). 

“Additionally, 30 healthy controls were included, more information on this study group can be found in S1 Appendix and S1 Fig.”

Here we describe their age, gender distribution and how they were processed. A sentence describing their recruitment has been added (lines 6-8).

“These controls were recruited among employees of the University of Antwerp and their families and friends.”

3.) The figure legends appear in the text but should be moved to the end.

As stated in the submission guidelines and confirmed by the editor in his communication the figure legends should be inserted in the text, immediately following the paragraph in which the figure is first cited. Therefore, we have not moved the figure legends.

4.) Results, page 15. “For the septic shock patients day 1 was used.” I presume this refers to the ROC for diagnostic biomarker. Please clarify and introduce the rationale for this in methods.

We have changed the title of Table 2 to “Cutoff values based upon the ROC curves of DPP4, FAP, PRCP and PREP” and have added the rationale for the use of day 1 in the Materials and Methods section describing the ROCs (lines 200-203).

“ROC curves were computed to assess the four enzymes as possible biomarkers for septic shock. For the septic shock patients (n = 40) only day 1 was used. The patient’s condition on day 1 is clinically the most relevant, since it is of utmost importance to identify patients as soon as possible to improve their outcome.”

5.) Discussion:” Since septic shock has a high mortality, another cutoff value was calculated in which no patients would be missed. In this case, each of the enzymes gives an insufficient specificity as more than half of the ICU control patients would be falsely diagnosed with septic shock.” This is not clear and requires better explanation.

In the Materials and Methods where we described the generation of the ROCs, we explain now how we defined a cutoff that would include all septic shock patients. This would mean that the sensitivity had to be 100% (lines 203-207). 

“ROC curves were generated using the ICU control group. Cutoff values were determined using two methods, one in which equal weight is given to false positives and false negatives, also called the Youden index, and another one in which all patients with sepsis are identified, meaning that the sensitivity reaches 100%.”

We have also decided to repeat this difference between the two methods in the Results section, so it is certainly clear for the reader (lines 278-282). 

“Cutoff values, either determined using the Youden index (giving equal weight to false positives and false negatives) or to obtain 100% sensitivity (to identify all patients with sepsis), are listed in Table 2 together with the sensitivity, specificity, negative and positive predictive value and the negative and positive likelihood ratios.”

In the discussion, we finally also added that we mean that the sensitivity had to be 100% (line and we also added that a lot of false positives would lead to unnecessary treatment (lines 396-400).

“Since septic shock has a high mortality, another cutoff value was calculated in which no patients would be missed (sensitivity is 100%). In this case, each of the enzymes gives an insufficient specificity as more than half of the ICU control patients would be falsely diagnosed with septic shock and would receive unnecessary treatment.”

6.) The Kaplan-Meier 90-day survival curves for the best PSE activity cutoffs should be shown as a figure in the main manuscript. The reader would like to see visually how PSE levels are associated with prognosis in septic shock.

Visualizing how PSE levels are associate with prognosis would technically be possible using Kaplan-Maier curves (splitting groups according to some cutoff for the PSE). Cutoffs could be based upon (for example) a ROC curve that uses PSE as predicting variable. However, when it comes to predicting the survival, the current data suffer from an ascertainment bias. Patients quickly recovering and leaving the ICU within one week, are not included. Patients who passed away within one week are also not included. As previously stated, we have included this as a weakness of the study (line 414-415). Therefore, although it would be technically possible to generate these curves, they may generate a biased view on the relation between PSE and survival at 90 days. Therefore, we chose not to show them.

References suggested by the reviewer:

1. Khanna A, English SW, Wang XS, Ham K, Tumlin J, Szerlip H, et al. Angiotensin II for the Treatment of Vasodilatory Shock. N Engl J Med. 2017;377(5):419-30.

2. Senatore F, Jagadeesh G, Rose M, Pillai VC, Hariharan S, Liu Q, et al. FDA Approval of Angiotensin II for the Treatment of Hypotension in Adults with Distributive Shock. Am J Cardiovasc Drugs. 2019;19(1):11-20.

3. Emdin M, Fatini C, Mirizzi G, Poletti R, Borrelli C, Prontera C, et al. Biomarkers of activation of renin-angiotensin-aldosterone system in heart failure: how useful, how feasible? Clin Chim Acta. 2015;443:85-93.

Reviewer #3: 

We would like to thank the reviewer for the careful revision. Please find our answer below.

The manuscript highlights markers that have the potential to become biomarkers for sepsis, for which currently no concrete association exists (as mentioned by authors as well). However, the associations described for most of the proline-specific enzymes were not very strong. As the authors mentioned further research is warranted. This manuscript will greatly benefit from more detailed analysis of age, gender, ethnicity to determine where the enzymes have any stronger correlation in any specific group. It is understandable data from human patients are not easily available however, to correlate the enzymes with progression in sepsis, this study requires control known inflammatory/anti-inflammatory patterns demonstrated side by side. This analysis will reveal the correlation of the inflammatory status, disease progression of sepsis 2 patient along with the enzymes can be potential biomarkers.

In our association study we included several (anti-) inflammatory parameters, such as IL-1β, IL-1RA, IL-6, IL-8, IL-10, TNFα, IFNγ and leucocytes. These were all the parameters regarding the immune system that were recorded for these 40 patients. As mentioned in the text (lines 353-355), we did not identify strong associations with the above mentioned parameters, except for PREP with IL-1RA. 

Further dividing our 40 patients into groups for a detailed analysis of age, gender or ethnicity will result in small groups that will not have enough statistical power and we have therefore opted not to include these analyses.

In future efforts other parameters can be included as well such as IL-17, IL-27, IL-33, which recently have been described to contribute to the immunological dysfunction in sepsis [5]. In addition, as we have mentioned in the manuscript, larger and more diverse patients should be recruited to validate the data presented here.

Regarding the diagnostic value (adjusted upon recommendation of reviewer 2) (lines 404-417):

“Therefore, in the future, it would be very interesting to study larger groups of patients and to prospectively include all patients at high risk of developing sepsis and measure, for example, their DPP4- and FAP-activities on a daily basis. … A limitation of this study is that only samples from patients who stayed in the ICU during the entire week, were available. Follow-up studies should include all patients diagnosed with sepsis. This may even increase the biomarker value of the enzymes studied here.”

Regarding the prognostic value (lines 430-431): 

“A stronger association might be identified when the DPP4-activity on day 1 is measured in a larger and more diverse sepsis patient population.”

References:

1. Shih C-J, Wu Y-L, Chao P-W, Kuo S-C, Yang C-Y, Li S-Y, et al. Association between use of oral anti-diabetic drugs and the risk of sepsis: a nested case-control study. Sci Rep. 2015;5:15260. 

2. Van Der Velden VH, Naber BA, Van Hal PT, Overbeek SE, Hoogsteden HC, Versnel MA. Peptidase activities in serum and bronchoalveolar lavage fluid from allergic asthmatics--comparison with healthy non-smokers and smokers and effects of inhaled glucocorticoids. Clin Exp Allergy. 1999;29(6):813–23. 

3. Lefebvre J, Murphey LJ, Hartert T V., Shan RJ, Simmons WH, Brown NJ. Dipeptidyl peptidase IV activity in patients with ACE-inhibitor-associated angioedema. Hypertension. 2002;39(2 II):460–4. 

4. Dellinger R, Levy M, Rhodes A. Surviving Sepsis Campaign: international guidelines for management of severe sepsis and septic shock: 2012. Crit Care Med. 2013;41(2):580–637. 

5. Morrow KN, Coopersmith CM, Ford ML. IL-17, IL-27, and IL-33: A Novel Axis Linked to Immunological Dysfunction During Sepsis. Front Immunol. 2019;10:1982.

---

## [Decision Letter · Decision Letter 1]

26 Mar 2020

Dysregulated activities of proline-specific enzymes in septic shock patients (sepsis-2)

PONE-D-19-30816R1

Dear Dr. De Meester,

We are pleased to inform you that your manuscript has been judged scientifically suitable for publication and will be formally accepted for publication once it complies with all outstanding technical requirements.

With kind regards,

Paul Proost, Ph.D.

Academic Editor

PLOS ONE

Additional Editor Comments (optional):

Reviewers' comments:

Reviewer's Responses to Questions

**Comments to the Author**

1. If the authors have adequately addressed your comments raised in a previous round of review and you feel that this manuscript is now acceptable for publication, you may indicate that here to bypass the “Comments to the Author” section, enter your conflict of interest statement in the “Confidential to Editor” section, and submit your "Accept" recommendation.

Reviewer #1: All comments have been addressed

2. Is the manuscript technically sound, and do the data support the conclusions?

Reviewer #1: Yes

3. Has the statistical analysis been performed appropriately and rigorously? 

Reviewer #1: Yes

4. Have the authors made all data underlying the findings in their manuscript fully available?

Reviewer #1: Yes

5. Is the manuscript presented in an intelligible fashion and written in standard English?

Reviewer #1: Yes

6. Review Comments to the Author

Reviewer #1: Thanks for addressing my concerns. My concerns have been resolved. I judged this revised manuscript 'accept'.

7. PLOS authors have the option to publish the peer review history of their article (what does this mean?). If published, this will include your full peer review and any attached files.

Reviewer #1: No

---

## [Editor Report · Acceptance letter]

7 Apr 2020

PONE-D-19-30816R1 

Dysregulated activities of proline-specific enzymes in septic shock patients (sepsis-2) 

Dear Dr. De Meester:

I am pleased to inform you that your manuscript has been deemed suitable for publication in PLOS ONE. Congratulations! Your manuscript is now with our production department. 

With kind regards,

on behalf of

Dr. Paul Proost 

Academic Editor

PLOS ONE